# Other models for data visualisations

*Paul Heinicker*

Other models of visualising aim at a (re)formulation of contemporary expectations and narratives concerning data and their visualisations as a very specific model of thinking data visualisation. It is precisely how and with what intention we work on and discuss visualisations that defines the conceptual space we open to this cultural technique. The concept of "other models" first points to the consequences and limitations of these ways of thinking. My positioning of the "other" consists first of the description of what it wants to distinguish itself from. I understand the "other visualising" as a chance to make the normative mode of data visualisation visible and discussable. In the discourse of visualisation, there is not yet an established language for critiquing the expectations of data images. The "other visualising" therefore establishes a negative way of reading the cultural and image phenomenon. As a first concretisation of these models, I formulate in the following a differently directed definition: data visualisation as intended violence.

# Data = Intention

Ideas and hopes around data visualisations are essentially oriented around two fundamental ideas of data visualisation: data and visualisation. With regard to data, I tend to describe contemporary data narratives using the figure of data exceptionalism as reproducers of a normative model of the imagination, practice, and reflection of data.

The concept of data exceptionalism enables to make visible a data positivist perspective, which is essentially defined by the rhetoric of the exception – the data phenomenon as a cultural turning point, a reductionist notion of data - solely numerical and technical, and a data forgetfulness in the sense of forgetting original – non-technical or mathematical approaches. A potential counter-position aims at broadening a narrowed notion of data, and this broadening has also been done by returning to existing concepts of data. Thus, in my perspective, it is primarily intentionality that characterises data. Data are not natural phenomena, but cultural artefacts of ordering structures. Data are not simply there, rather they are intentional. They are created from a particular perspective, in an artificial process, and for an application or reception. This data intention can be concretised in the reflection of the models that produce this data. Thus, at least two model applications are found in the intentional use of data. On the one hand, data - defined by me as abstractions - are not to be understood as images of reality, but as conscious projections of one or more models about this reality. On the other hand, I also understand the various modes of data practices as models applied with a purpose. Data exceptionalism is then understood as dealing with data in a particular model, namely in a positivist way. The ideas and intentions about what can be considered or produced as data and how to work with data are primarily shaped by models.

Probably the most important insight that comes from considering data exceptionalism is the aspect of modelling. The added value of data does not lie in the longed-for automated analysis of patterns in them, but more tellingly in the reflection of the models they produce. Data are both mirrors and producers of social reality. From this perspective, data are not the cause of social asymmetries, but rather an effect of a particular conception of what to do with the data. Data exceptionalism then only describes a certain model to proceed in a data positivist way. The questions about this model, i.e. questions why and for what purpose data is used, then promises possibly even more epistemic value than the analysis of the data itself. What is needed, according to this line of reasoning, is not another algorithmic, computational, or digital turn, but a return to the ideas, notions, and concepts, in short, the modelling of data. Data, by definition, are understood as abstractions, not images of reality, but always projections of a model about that reality. The deficiency of data is not that they are reduced in capacity, but that the confidence of completeness is ascribed to them by society.

# Visualisation = Violence

In relation to the object of visualisation, I distinguish the practice of visualisation in two central forms. In a dichotomous arrangement, I differentiate affirmative and, opposite to that, critical approaches. "Affirmative" I interpret as an attitude toward the data to be visualised that takes them as given and their visualisation as unqualifiedly necessary. Instead of this efficiency- and optimization-driven idea of an image-driven visibility of data, more agile concepts or models should be found that can grasp the process of visualisation more profoundly in terms of its epistemic potential. What is problematised with this conceptual "immobility" is the tendency of the affirmative visualisation model to seem hopeless. Visualisation should rather be understood in its transformative processes, which independently of the object design their own reality and thus their own knowledge, which needs to be reflected accordingly. Therefore, alternative models are needed that attempt to describe the limits and possibilities of the cultural technique of visualisation.

   In this context, my ideal of the "other visualising" also concretises itself. The "other" means approaches to the idea of visualisation that, apart from the affirmative visualisation models, is based on the critical reflection of the underlying models of thought. In addition to the critique of established conventions, it is primarily a diagrammatic position that understands visualisations as a projection of models. In contrast to a passive understanding of visualised diagrams as a rigid and (re)clarifying order, the diagrammatic is thought of as an active process that designs new arrangements or models in the relation of structures. What unites all these diagrammatics is that they push a certain structure through the filter of a conceptual model or world order onto its object. It is the purposeful transformation of data into a particular order that can be described as violent. Thus, again, there are at least two types of models that shape the process of visualisations. First, it is the notion of how visualisations are conceived: as an affirmative form of legible visualisation, the structural reading as diagrammatic reordering, or even the cosmogrammatic projection. Second, it is then the violent transformation of a data base, shaped via a particular model, that can result in any number of visualisations, depending on which model is chosen.

# Data Visualisation = Intended Violence

As a consequence, I understand data visualisations in their intentional and enforced implementation as intended violence. Data is abstracted from an arbitrary object through a particular model, and then in turn made perceptible through the model of a transformation. In this double model arrangement, the relational aspect of visualisations becomes clear, inscribing itself as a process of projection. Data visualisations do not represent, but rather design their very own images in a cascading transformation of structures. The interpretive directions of this insight are, however, open. A designer or recipient of a visualisation can open up to this circumstance, but these phenomena function intrinsically without this awareness. The model perspective on visualisations is only one possible form of critical questioning. However, it enables diverse moments of insight.

   Other models are ultimately intended to give indications of how visualisations are to be conceived as a cultural technique. The goal is not the search for the one visualisation that is to be optimised ever further in its readability and mediation efficiency. Rather, of relevance is an inefficiency that can allow and open up the diversity and complexity of visualisation culture. Instead of the contemporary culture of exclusion by a dominant (and affirmative) model, ideas that deviate from it should also be involved in the creation of visualisations.

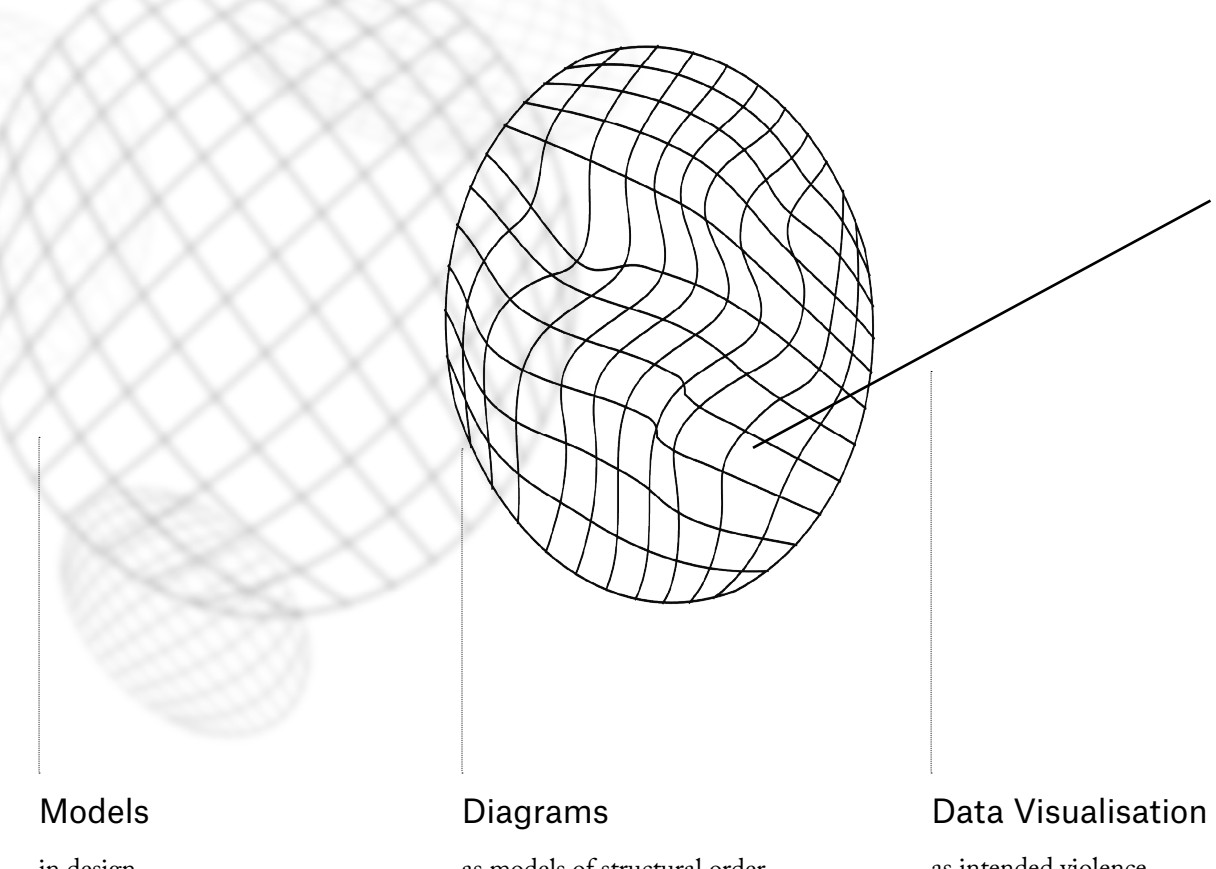

Models

in design

Diagrams

as models of structural order

Data Visualisation

as intended violence