# OpenReview forum: "Other models for data visualisations"
_IEEE.org/2022/Workshop/altVIS — Reject_

### Official Review · Reviewer_XqNz · 2022-08-10

**Review:**

In this essay it is argued that data visualization is "intended violence", a provocative stance which is unpacked by providing a meaning to data as being an artifact of intention and a meaning to visualization as being violence. In doing so it is meant to manifest itself as an "otherized" theory such that normative visualization might be better highlighted (a lofty goal that I'd be excited to see executed!). Yet it feels, to me, that it falls short of reaching such a conclusion and does not successfully locate where normative visualization exists in all but the most superficial manner, that ignores the good works continually taken on by our community. Compounding this opinion is that, I confess, I had quite a difficult time actually reading this paper. It is quite dense, often does not explain its terms, or provide reference to relevant literature which might help to further the its points (or even just meaningfully locate itself within prior scholarship). Perhaps with clearer slower discussion, this provocation might arrive at something great.

Alternatively: clearly I am provoked by this provocation, as this review has now gotten to be 769 words, (nearly two thirds of the length of the essay itself) so maybe that's value enough.

In any case, I want to respond to a few points:

1. The first point that data=intention or that data is never raw, is one made repeatedly in other places (cf "data feminism", "Raw Data" Is an Oxymoron", up/downstream references made in/to those works, etc). While it is something valuable to highlight, it is perhaps said more clearly and concisely by just locating among prior work. Relatedly: i believe that "intended violence" is the incorrect working of this conjunction of this idea with the next, it should be _intentioned_ violence.

2. The next point identifies visualization as being violence. It does so by identifying visualization as an act of transformation rather than a thing unto itself. This is a point I _strongly_ agree with: it is a core observation motivating algebraic visualization design, a theory to which my publication record would attest that I am a strong proponent. This observation is valuable and has lots of great consequences and conclusion, however I do not believe one is that such transformations can be identified as violence. I argue extreme transformation is not violence; a seed transforms into a tree in a cautious collaborative manner that (at least naturally) is in dialogue with its surroundings (cf "Hidden Life of Trees"); humans transform from young into old not through violence but through maturation. Perhaps it is human mediated transformation then that is to be thought of as violence? I'd argue against this as well: paints and canvas (or cave walls or whatever) are transformed into works of art. While it could be argued any use of resources must itself require "violence" (eg the materials in the paints taken from the earth), the point then would be that any human action is violence and identifying only transformation as violence is more akin to cherry picking.

3. I printed the PDF and found that the figure on the fourth page included a large black rectangle directly over the models component (which ostensibly is an artifact of my printer handling the opacity in the figure poorly), however I found that this printing error might serendipitously dovetail with some of the form of the text. In particular the notion of what model is discussed but not defined, referenced without being presented directly, much in the same way that the model interacted with the rest of the figure as rendered with the unintended black square.

4. A stance that is briefly taken is that the visualization community is unwilling to embrace alternative ideas and models of the process of data visualization, and is fully beholden to concepts of efficiency and mechanistic utility. I believe that this is a shallow reading that ignores numerous lines of work, for instance Hullman and co's "Benefitting infovis with visual difficulties", Brüggemann and cos "The fold: Rethinking interactivity in data visualization", Offenhuber's notion of autographic visualizations, this workshop, and just so many other great works. It seems that this point may be intended to serve a rhetorical defense (something like "rejecting this work would be evidence of non-acceptance of alt.models"), however I think that existence of a vibrant heterogenous academic community that does actively consider and explores a wide variety of alternative models to pure utility does this reflexively defense point in.

(I put forward these notes not to, uh, violently attack this paper but as a response in kind to the work.)

**Conflicts:**

None!

**Review Inclusion:**

Yes

**Sufficiently Alt:**

Yes

**Superlative:**

Most Inscrutable

---

### Official Review · Reviewer_99hV · 2022-08-15

**Review:**

Either I'm not smart enough to comprehend the meaning of this work, or it is meant to be some strange prank from the author.

**Conflicts:**

None.

**Review Inclusion:**

No

**Sufficiently Alt:**

Yes

**Superlative:**

Most incomprehensible.

---

### Official Review · Reviewer_ue86 · 2022-08-24

**Review:**

The paper doesn't even satisfy the basic formatting requirement and is definitely not ready for publications. Should the paper be desk rejected?

**Conflicts:**

NA

**Review Inclusion:**

No

**Sufficiently Alt:**

No

**Superlative:**

Sloppiest

---

### Official Review · Reviewer_wH7L · 2022-08-31

**Review:**

In this paper, the author proposes a reflection on data visualisation and how it can be considered as "Intended Violence".

The paper in itself is not easy to follow as it introduced quite a lot of concepts without clearly defining them. From what I understand, the visualisation domain tends to consider data visualisation as an accurate representation of the data and what it represents. Such representation is considered as a ground truth by the viewer and it could be considered as a truth imposed on the viewer, hence the term "Intended Violence" used by the author. Data themselves are actually just a partial view of a model (most of the time too complex to be described mathematically). The author then called for visualisation models that would propose a less imposed visualisation and try to propose visualisation as an interpretation instead of an affirmative truth.

The ideas are very interesting and really deserved to be exposed and discussed in the visualisation community. However, as I mentioned before, I am not sure to completely understand what is in the paper, which is an issue and might be a sign that it is too complex to be understood as is by a researcher in visualisation (I may be wrong and I may be the only one not to understand it). The important concepts should be defined clearly, with examples, and maybe some are not necessary (like "data exceptionalism" for instance). Examples should also be included and explained to illustrate the main concept of the paper: visualisation = intended violence.

Finally, I wonder if some visualisation concepts already touch upon such reflection. We could consider the whole field of uncertainty visualisation for instance.


**Conflicts:**

No Conflict

**Review Inclusion:**

No

**Sufficiently Alt:**

Yes

**Superlative:**

Most abstract

---

### Official Review · Reviewer_B7YH · 2022-08-31

**Review:**

As your meta review, I want to start off by saying your paper provoked a lot of discussion on and offline. As a program committee we have decided to extend a conditional acceptance to the paper. There are several points addressed in two of the reviews below that I will summarize more broadly here. We ask that you revise the paper by:
- Referencing prior work
- Make the language and prose legible to VIS audiences

Our main concern was the legibility of the paper and its lack of acknowledgment of prior and related ideas. We look forward to the revisions -- submitted by 30 Sept. If you would like clarification or will not be able to complete the changes in this timeframe, please write to the OC (at alt.vis.workshop@gmail.com).

**Conflicts:**

None that I know

**Review Inclusion:**

No

**Sufficiently Alt:**

Yes

**Superlative:**

Most controversial

---

### Decision · Program_Chairs · 2022-08-31

Reject